

# Research overview on the genetic mechanism underlying the biosynthesis of polysaccharide in tuber plants

Mengwei Xu[1], Jiao Hu[1], Hongwei Li[1], Kunqian Li[1] and Delin Xu[1,2]

[1] Department of Medical Instrumental Analysis, Zunyi Medical University, Zunyi, Guizhou, China
[2] Guizhou Provincial Demonstration Center of Basic Medical Experimental Teaching, Zunyi Medical University, Zunyi, Guizhou, China

## ABSTRACT

Tuber plants are of great significance in the world as human food crops. Polysaccharides, important metabolites in tuber plants, also serve as a source of innovative drugs with significant pharmacological effects. These drugs are particularly known for their immunomodulation and antitumor properties. To fully exploit the potential value of tuber plant polysaccharides and establish a synthetic system for their targeted synthesis, it is crucial to dissect their metabolic processes and genetic regulatory mechanisms. In this article, we provide a comprehensive summary of the basic pathways involved in the synthesis of various types of tuber plant polysaccharides. We also outline the key research progress that has been made in this area in recent years. We classify the main types and functions of tuber plant polysaccharides and analyze the biosynthetic processes and genetic regulation mechanisms of key enzymes involved in the metabolic pathways of starch, cellulose, pectin, and fructan in tuber plants. We have identified hexokinase and glycosyltransferase as the key enzymes involved in the polysaccharide synthesis process. By elucidating the synthesis pathway of polysaccharides in tuber plants and understanding the underlying mechanism of action of key enzymes in the metabolic pathway, we can provide a theoretical framework for enhancing the yield of polysaccharides and other metabolites in plant culture cells. This will ultimately lead to increased production efficiency.

## INTRODUCTION

Polysaccharides play a vital role in the growth and development of tuber plants. They can be found in the form of starch, cellulose, and pectin, and are crucial for the morphology, growth, development, and defense of tuber plants. These sugar chains are formed by glycosidic bonds and consist of more than 10 monosaccharides. Homo-polysaccharides, like starch and cellulose, are made up of the same monosaccharide, while heteropolysaccharides, such as gum arabic, contain different monosaccharides. Polysaccharides are glycosides and can be hydrolyzed to produce intermediate products. In recent years, natural polysaccharides have shown a wide range of pharmacological properties, including anti-tumor, immunomodulatory, antioxidant, and anti-inflammatory effects (*Wang,*

Corresponding author
Delin Xu, xudelin2000@163.com

*Kanyuka & Papp-Rupar, 2023*). Polysaccharides derived from tuber plants not only have high nutritional value and energy storage capacity, but also play a vital role in the prevention and treatment of various diseases. Increasing the yield of polysaccharides can be achieved through various methods, such as overexpressing key genes in polysaccharide biosynthesis, co-expressing multiple genes, directed evolution of enzymes, and blocking polysaccharide anabolic bypass (*Yang, Yang & Zhang, 2020*). Therefore, a comprehensive understanding of the genetic mechanisms involved in polysaccharide biosynthesis in tuber plants is crucial for their widespread application in diverse fields, including food and beverage production, drug development, and the creation of new materials.

The biosynthesis of polysaccharides is a complex process that is regulated by a network of proteins. Recent studies have identified various genes and enzymes responsible for regulating different pathways of polysaccharide biosynthesis in tuber plants. For example, genes involved in producing different starch components, such as amylopectin and amylose, have been identified in potato tubers (*Nakamura et al., 2022*). Enzymes involved in the synthesis of tuber starch have also been characterized. Furthermore, research has focused on investigating the regulatory mechanisms involved in the process of tuber development, leading to the identification of genes responsible for the biosynthesis and storage of soluble sugars and the regulation of enzyme activity in the process. Genetic mechanisms involved in the synthesis of other polysaccharides, which are present in some tuber plants like *Jerusalem artichoke*, have also been explored (*Sawicka et al., 2020*).

In recent years, numerous studies have focused on the mechanisms of enzymes and genes related to specific plants or the metabolism of single polysaccharides. However, there have been few reports summarizing a specific classification of plants known as tubers. This lack of targeted summaries has resulted in a dearth of research on the biosynthetic mechanisms of the main polysaccharides found in tuber plants. Tuber plants hold a significant position in the field which are an essential component of land plants (*Dong et al., 2021*). This article aims to address these gaps by summarizing the main types and functional properties of polysaccharides in tubers. Additionally, it will describe the biosynthetic pathways and the processes of key synthases. The study also emphasizes the potential for industrial exploitation and clinical application of polysaccharides. Furthermore, the article concludes with an examination of the genetic mechanisms involved in regulating polysaccharide synthesis in tuber plants. This comprehensive analysis seeks to establish a foundation for further research on the regulation of polysaccharide synthesis in plants, as well as provide a theoretical basis for the use of directed synthesis to improve polysaccharide yields in industrial production.

For survey methodology of this article, we firstly surveyed the related articles of the past 3–5 years by searching the key words of tuber plants, polysaccharides, biosynthetic, genetic mechanism and metabolic pathway in PubMed, Web of Science, NCBI, Baidu Scholar and Bing Scholar. Then we concluded the valued information for preparing this article.

## APPLICATION VALUE OF THE TUBER PLANT POLYSACCHARIDES

Tuber plants contain various types of polysaccharides, including starch, cellulose, pectin, gum, mucilage, and fructan, which serve different purposes and demonstrate medicinal properties (Table 1). Polysaccharides have numerous applications and values. Firstly, their physicochemical characteristics make them suitable for the production of pharmaceutical materials and drug releasers (Zierer et al., 2021). Some polysaccharides have properties such as easy gel formation, high osmotic pressure, high viscosity, and excellent water absorption (Dong et al., 2021). Secondly, their pharmacological properties, antigenicity, anti-tumor effects, and other biological functions make them useful for the development of new drugs or vaccines (Cheng et al., 2023). For example, yams' starch and mucopolysaccharides help prevent cardiovascular disease, diabetes, and intestinal microbiota disorders (Epping & Laibach, 2020). Additionally, the polysaccharides and other components in Salvia exhibit antioxidant pharmacological activity (Fu et al., 2023). The polysaccharide components of Ginseng, Atractylodes macrocephala, Poria, and Licorice are used to treat neuroendocrine disorders, gastrointestinal motility, and hormonal abnormalities (Ma et al., 2021). Bletilla striata's yeast polysaccharide has pharmacological effects such as wound healing, hemostasis, antioxidant, anti-inflammatory, anti-fibrotic, and in vitro immunomodulatory activities (Xu et al., 2021). Furthermore, the dried tuber of Bletilla striata is an important astringent and hemostatic drug widely used for treating gastrointestinal mucosal injuries, ulcers, bleeding, bruises, and burns (Jiang et al., 2021). The Polygonatum sibiricum tuber's polysaccharide exhibits antioxidant, antibacterial, and antitumor properties while also lowering blood sugar and lipids, enhancing the immune system, and displaying remarkable moisturizing and moisture-proof properties (Yu et al., 2022). Gastrodia elata's dried tuber provides relief from gout, spasms, and calms liver fire. It is rich in polysaccharide components with anti-aging, anti-tumor, immunomodulatory, and other functions (Lu et al., 2022).

## METABOLIC PATHWAY SYNTHESIS OF POLYSACCHARIDES IN TUBER PLANTS

Sugars play a crucial role in the metabolism of plants, providing them with the energy and essential intermediate products necessary for survival, growth, development, and reproduction. Through the process of photosynthesis, green plants convert carbon dioxide and water into sugars. These sugars can then be further metabolized into vital substances such as adenosine triphosphate (ATP), coenzyme (NADH), pyruvate, phosphoenolpyruvate, erythrose-4-phosphate, ribose, and various other compounds. These substances are vital for maintaining the life processes of plant organisms and are synthesized through different metabolic pathways (Duan et al., 2021).

For the synthesis of other polysaccharides as the end product of photosynthesis, sucrose must either be hydrolyzed into glucose and fructose using converting enzymes or catalyzed into fructose through sucrose synthase (Fig. 1). Before they can be utilized by the biosynthetic enzymes and glycosyltransferases (GTs) in the cell, monosaccharides

Xu et al. (2024), *PeerJ*, DOI 10.7717/peerj.17052

**Table 1  Main classification and functional application of polysaccharides in plant tubers.**

| Polysaccharide classification | Main storage areas | Functions | Physical and chemical characteristics | Medicinal properties | Reference |
|---|---|---|---|---|---|
| Starch | Seed and tuber | Plant nutrients, food industry, bioactive ingredient delivery vehicles | Polysaccharide polymer compounds composed of glucose, including amylose and amylopectin, with different adsorption properties depending on the molecular form. Present white powder state, insoluble in cold water, soluble in hot water, easy to paste in hot water, aging when the temperature decreases | Natural starches separated from natural sources are used as adhesives and disintegrators in tablet formulations. In the formulation of metronidazole tablets, the natural starch in the sedge tuber is used as a binder. | *Chakraborty, Kalita & Sen (2019), Qiu et al. (2023), Sun et al. (2023)* |
| Cellulose | Cell wall | Provide cohesion, protection and directional growth for plants with stability. | D-glucopyranosyl groups insoluble in water, dilute acids and bases at room temperature are chain-like macromolecular compounds linked by β-1,4 glycosidic bonds. | The addition of nanocellulose polysaccharide can improve the antibacterial activity of polysaccharide. Bacterial cellulose has been used as a source for human drug delivery systems, and skin patches based on bacterial cellulose are used to load different drug molecules; Formulations of tizanidine (water soluble) and Famotidine (less water soluble) oral tablets using bacterial cellulose as the sole excipient, both drugs in capsules have excellent release effects. Bacterial cellulose as a laxative medicine has used a variety of hygroscopic. | *Ul-Islam et al. (2020), Niu et al. (2023), Pedersen et al. (2023)* |
| Pectin | Cell wall and intercellular layer | It regulates plant adaptation to low temperature photosynthesis, regulates plant sucrose distribution, promote intercellular adhesion, provide structural support in the main wall; influence the secondary wall formation of fiber and woody tissue; provide a reservoir of oligosaccharide signaling molecules important for plant growth and defense responses, affects wall rheology and supports seed and root growth. | Include homogalacturonan (HG), rhamnogalacturonan I (RGI) and rhamnogalacturonan II (RGII), different pectin domains are covalently linked to each other in the cell wall and have great gelation, emulsification and antioxidant properties. It is a soluble dietary fiber with good gelation, emulsification and antioxidant properties. | The physicochemical and antibacterial properties of pectin can be used as multi-functional pharmaceutical excipients and nutritional products. Pectin and its modified nanocomposites can be used for pharmaceutical and drug delivery. | *Kedir, Deresa & Diriba (2022), Wang et al. (2022)* |
| Gum | All organizations | Aqueous dispersions usually have the characteristics of suspension, dispersion, emulsification adhesive or viscous, and gel, commonly used as coagulants, adhesives, lubricants or film-forming substances, biocompatible, biodegradable, stable, and will not cause immune response | As a water-soluble polysaccharide substance, the aqueous dispersion of gum usually has the characteristics of suspension, dispersion, emulsification gum or viscous, and gel, *etc.* It is often used as coagulant, adhesive, lubricant or film-forming substance. | Gum can be used as an excipient, replacing Astragalus gum and starch as a binder and disintegrator, respectively; used in the development of biomaterials, which can be used in surgery, as a cell growth scaffold during tissue regeneration, gum scaffolds have good support for the growth of mesenchymal stem cells (MSC); be used as a component of synthetic substrates for therapy and tissue repair. | *Amaral et al. (2022), Owusu et al. (2022), Tuteja & Nagpal (2023)* |

Xu et al. (2024), *PeerJ*, DOI 10.7717/peerj.17052

**Table 1** (*continued*)

| Polysaccharide classification | Main storage areas | Functions | Physical and chemical characteristics | Medicinal properties | Reference |
|---|---|---|---|---|---|
| Mucilage | Mucus cells of thin-walled tissues | It has strong hygroscopicity, swells rapidly in water, dissolves to form viscous slurry, and is insoluble in organic solvents. It is often used as lubricant, suspension agent and auxiliary emulsifier in medicine. | It is mainly a compound of high molecular weight polymeric polysaccharides combined with organic acids. It contains galactose, pentose and methyl pentose, which are linked to glyoxylate residues via glycosidic bonds. Due to its anionic structure, the mucilage can interact with other cationic polymers, resulting in the formation of polyelectrolyte compounds. | The mucilage is suitable for formulating uncoated tablets and can be safely used as a drug excipient. The flow characteristics of the particles prepared with mucilage have good compressibility compared with those prepared with starch and PVP. | *Ilango et al. (2022), Xu, Hu & Li (2023), Goksen et al. (2023)* |
| Fructan | Roots, stems, leaves and seeds of Asteraceae, Gramineae and Liliaceae | As an autotrophic and heterotrophic organ of plants, it regulates the adaptation of plants to low temperature photosynthesis, regulates the distribution of sucrose in plants and adapts plants to water deficit, and is often used in the production of fructose after hydrolysis. | Polymorph of $\beta$-D-fructose, white powder. Soluble in water, low viscosity of aqueous solution, similar to the properties of gum Arabic. Insoluble in more than 65% ethanol. With dextrose. | Fructan, as a prebiotic, can selectively promote the activity and growth of specific native bacteria to regulate the gut flora in patients with inflammatory bowel disease (IBD). The addition of fructan to food can significantly increase the number of bifidobacteria, thereby improving the gut microbiome, and its fermentation products can reduce the gut pH and inhibit the growth of pathogenic and spoilage bacteria. In addition, fructooligosaccharides may reduce the risk of colon cancer. The production of butyrate from fructan by intestinal flora can reduce the incidence of tumor and inhibit tumor growth and metastasis. | *Wan et al. (2020), Dosio et al. (2023)* |

Peer

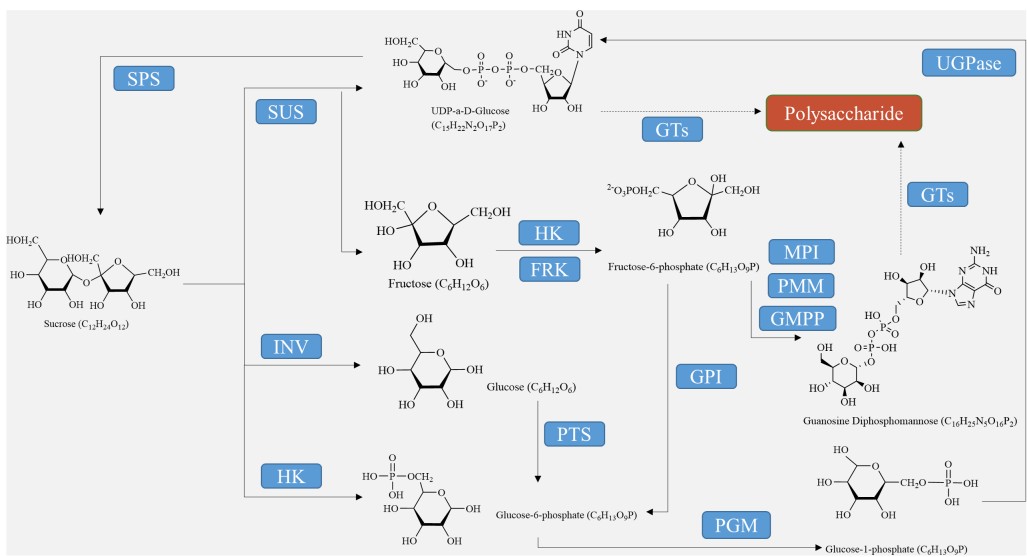

**Figure 1** Polysaccharide synthesis process regulated by enzymes encoded by structural genes.

need to be activated as NMP or NDP derivatives. Examples of activated monosaccharides include ADP (adenosine diphosphate), TDP (thymidine diphosphate), GDP (guanosine diphosphate), and UDP (uridine diphosphate) (*Thibodeaux, Melançon & Liu, 2007*). Glucose-6-phosphate and fructose-6-phosphate are the primary sources of nucleotide sugars (Fig. 1). In the biosynthesis of GDP sugars, phosphomannoisomerase (PMI) converts fructose-6-phosphate to mannose-6-phosphate (*Tanaka et al., 2023*). Conversely, glucosamine-6-phosphate synthase (GlmS) converts fructose-6-phosphate to glucosamine-6-phosphate for the formation of UDP sugars. Alternatively, UDP-sugars can be produced from galactose through the Leloir pathway, ultimately leading to UDP-glucose (*Thibodeaux, Melançon & Liu, 2008*). Although glucose-6-phosphate serves as a precursor for many UDP-sugars, it is more commonly utilized in the biosynthesis of TDP- and CDP-sugars (*Hua et al., 2021*). Finally, monosaccharides are enzymatically catalyzed by various GTs to bind together and form growing polysaccharide polymers, which are then polymerized and exported to create plant polysaccharides (*Wingler & Henriques, 2022*). These substances are further metabolized, eventually resulting in the formation of polysaccharides such as starch, cellulose, pectin, fructan, and other metabolites at different sites through enzymatic catalysis (*Lynch, 2022*).

## Metabolic synthesis pathway of starch

In plant tubers, the main sites for starch synthesis are chloroplasts and amyloplasts. Within the chloroplast stroma, the Calvin cycle utilizes $CO_2$, which enters the chloroplast matrix through the stomata (*Lee et al., 2022*). Initially, it forms 3-phosphoglycerate (3PG) by reacting with its receptor RuBP under the influence of Rubisco carboxylase. Subsequently, 3PG is reduced to glyceraldehyde-3-phosphate (G3P) (*Sharkey, 2023*). G3P serves two purposes: some is transported to the cytosol to produce sucrose *via* a series

**Figure 2  Starch synthesis pathways in chloroplasts.**

of biochemical reactions, while the rest is converted to fructose-6-phosphate (F6P). F6P is further transformed into ADP-glucose (ADP-Glucose) through glucose-6-phosphate (G6P) and glucose-1-phosphate (G1P), ultimately culminating in starch production (*Funfgeld et al., 2022*). On the other hand, in amyloplasts, starch is produced using sucrose, generated during photosynthesis, as a carbon source. Sucrose is conveyed from the leaves to the seeds, where it is hydrolyzed by either sucrose synthase or sucrose convertase to produce fructose and ADP-glucose (*Liu et al., 2022*). ADP-Glucose is then converted by ADP glucose pyrophosphorylase (AGPase) into hexose phosphate (G1P and G6P), leading to further ADP-glucose synthesis (Fig. 2) (*Zhu et al., 2022*). Finally, ADP-glucose produces amylopectin through the activities of starch isomerase, starch branching enzyme, and soluble starch synthase, while another portion is transformed into amylose by granule-bound starch synthase and stored in endosperm cells (*Lee et al., 2022*). In sugar metabolism, sucrose phosphate synthase (SPS) and sucrose phosphatase (SPP) are present as compounds in plants, and the process of sucrose production catalyzed by SPS is irreversible (*Worden et al., 2015*; *Mason et al., 2023*).

## Biosynthesis pathway of cellulose

Cellulose is synthesized directly at the plasma membrane in plants. The remaining portion of the wall, consisting of hemicellulose and pectin, is formed within the Golgi apparatus and released into the apoplast through exocytosis (*Verma et al., 2023*). Bacteria have the ability to produce a variety of polysaccharides, which are generated and transported by different membrane protein complexes. Similarly, chitin and glucan chains in fungi are synthesized directly at the plasma membrane by glycosyltransferases, while maintaining the same meaning (*Noack & Persson, 2023*). Cellulose synthesis involves a complex of multiple

**Figure 3** Cellulose biosynthesis pathways.

cellulose synthases. Cellulose synthase complexes (CSCs) are assembled in the Golgi and then transported to the cytoplasmic membrane for synthesis. The CSCs consist of a rosette structure composed of six subunits, each containing six cellulose synthase monomers. The arrangement of microfibrils within the cell wall is influenced by cellular microtubules and microtubule dynamin/kinesin, and cell wall proteins also play a role in this process (*Wang et al., 2020*). Furthermore, certain membrane proteins, such as the Kor protein, participate in cellulose synthesis (*Hoffmann et al., 2021*).

The movement of CSCs along cortical microtubule pathways defines the direction of cellulose microfibril synthesis (*Chebli & Geitmann, 2023*). The regulation of cellulose biosynthesis includes the transcriptional regulation of CesA genes, post-translational modification of CesA proteins, assembly, transportation, and localization of CSC complexes, as well as the regulation of other enzymes involved in glucan synthesis (Fig. 3) (*Wang et al., 2022*). The circulation of CSCs between the plasma membrane and various intracellular compartments plays a crucial role in determining the level of cellulose synthesis (*Vellosillo et al., 2021*). Studies have demonstrated that several factors, such as transcription factors like MYB family proteins, SND1, VND family proteins, and signaling molecules including NO, NAA, and BR, can promote cellulose synthesis (*Guo et al., 2022*).

## Pectin biosynthesis pathway

Pectin, which is a crucial component of the plant cell wall, is primarily composed of three essential polysaccharide structural domains: homogalacturonic acid (HG), rhamnogalacturonan I (RG-I), and rhamnogalacturonan II (RG-II) (Fig. 4). Homogalacturonan, the simplest structure of pectin, consists of 1,4-linked acidic residues of α-D-galacturonic acid (GalA). Some of these residues have a methyl esterified carboxylic group (*Zdunek, Pieczywek & Cybulska, 2021*). RG-I is composed of alternating disaccharide units of 4-linked α-D-galacturonic acid and 2-linked α-D-rhamnose. It also contains branches with side chains of arabinan, galactan, and arabinogalactan. On the other hand, RG-II is the most complex form of pectin, with high substitutions of monosaccharide residues such as rhamnose, xylose, arabinose, galactose, and some deoxy sugars like aceric acid, KDO, and DHA. These residues are linked together by 20 different glycosidic linkages (*Kumar et al., 2023*).

Pectins undergo polymerization, methyl-esterification, and modification in Golgi stacks before being transported to the cell wall in highly methyl-esterified forms. The synthesis of pectin requires various enzymes, including glycosyltransferase, methyl-transferase,

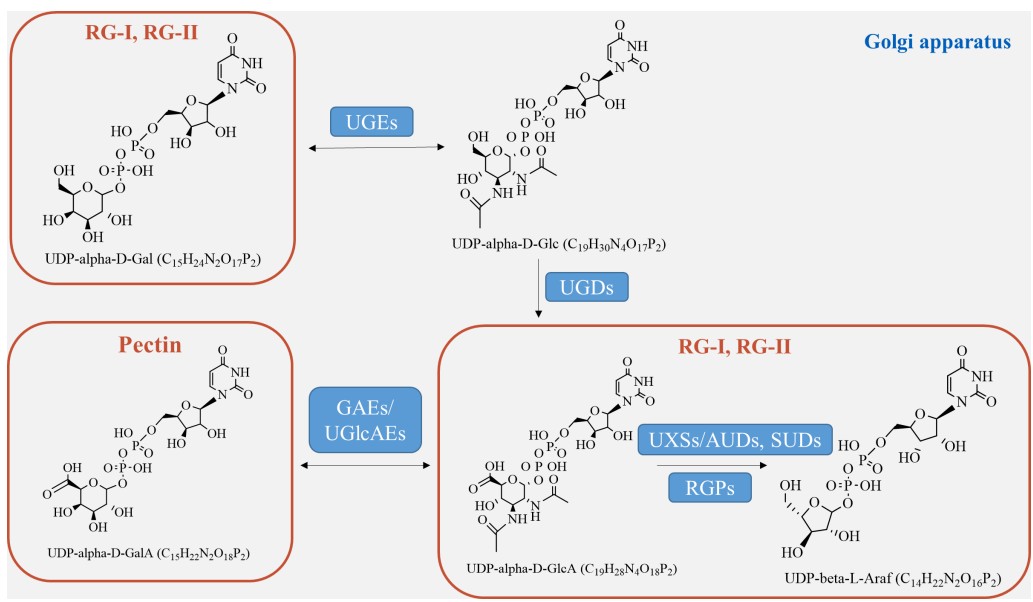

**Figure 4** **Pectin biosynthesis pathways.**

and acetyltransferase, which are involved in constructing complex structures of pectin polysaccharides in the cis-Golgi, intermediate Golgi, and trans-Golgi (*Li et al., 2023*). Highly esterified pectins are modified by pectin-degrading enzymes, such as pectin methylesterases (PME), through demethylesterification and deacetylation processes, resulting in the production of pectate and the release of methanol (*Kumar et al., 2023*). Pectin biosynthesis relies on different nucleotide sugars as active sugar donors, which are converted from free sugars into nucleotide sugars by a few nucleotide sugar precursors (UDP-Glc, GDP-Man). This conversion process involves reciprocal enzyme actions followed by a series of sequential reactions catalyzed by glucokinase and UDP-glucose pyrophosphorylase (*Tan et al., 2022*).

The formation of the intine can be regulated by certain genes involved in pectin metabolism (Fig. 4). UDP-sugar pyrophosphorylase (USP) is involved in the processes of synthesis and modification of the cell wall during pectin synthesis, as pectin is the main component of the intine layer (*Liu et al., 2021*). Deletion occurs in the USP mutant due to the impaired synthesis of pollen intine (*Mi et al., 2022*). *Pectic ArabinoGalactan synthesis-Related (PAGR)* encodes a protein belonging to the DUF-246 family. *PAGR* can influence the formation of AGPs and the structure of RG-1 (*Stonebloom et al., 2016*). Abnormal pollen germination occurs as a result of the mutant's affected intine, which in turn impacts pectin synthesis (*Ma et al., 2021*). Moreover, the modification of pectin side-chains by specific genes plays a crucial role in pollen development, germination, and pollen tube growth (*Zhou et al., 2022*). In the presence of $Ca^{2+}$, pectin degradation occurs through β-elimination catalyzed by *PLL (pectate lyase-like)* (*Safran et al., 2023*). Demethylesterified pectin can be degraded catalytically by polygalacturonase (PG), either as an internal or external enzyme (*Olawuyi et al., 2022*). PMEs play a role in the development of pollen

and pollen tubes by promoting the demethylesterification of pectin (*Ma et al., 2021*). In *Arabidopsis*, specific genes expressed in pollen grains or pollen tubes, namely *VDG1* and *PPME1*, have been identified and are involved in maintaining the homeostasis of pollen tube growth (*Tang et al., 2023*). Additionally, *PME48* regulates pollen germination and affects the reconstruction of pollen intine (*Ma et al., 2021*). There is a delay in pollen germination and a significant decrease in the germination rate in the *PME48* mutant, accompanied by the production of two pollen tubes from some pollen grains. The development of intine, pollen germination, and pollen tube growth are regulated by *PMEI*s, which inhibit the activity of *PME*s by binding to them (*Kim et al., 2020*). In summary, these factors collectively play a crucial role in the development of pectin biosynthesis.

## Biosynthetic pathway of fructan

The biosynthesis of fructan in plants utilizes sucrose as a donor of fructose. The plant glycoside hydrolase family *GH32* consists of both synthetases and degrading enzymes, collectively known as fructan active enzymes (FAZYs). These FAZYs are responsible for regulating the diversity of structure and size of fructans (*Hu et al., 2022*). Fructan biosynthesis occurs in vacuoles and is catalyzed by two or more fructan transferases (FTs), which transfer a portion of fructose from the donor substrate to the acceptor molecule (*Lekakarn et al., 2022*). On the other hand, fructan hydrolysis is facilitated by fructan exohydrolase (FEH), which cleaves fructose residues located at the terminal end (*Matros et al., 2021*). Unlike other polysaccharides, fructan synthesis does not require phosphorylation or nucleotide cofactors. Instead, various combinations of FTs engage sucrose in the synthesis of different types of fructans (*Chen et al., 2023*). In a qualitative study on fructan metabolism in *Helianthus tuberosus* tubers, it was found that the entire metabolic process of inulin-type fructan involves two enzymes: sucrose: sucrose 1-fructosyltransferase (1-SST) and fructan: fructan 1-fructosyltransferase (1-FFT) (*Huang et al., 2021*). In the presence of 1-SST, sucrose undergoes an irreversible one-step transfer of fructose from one sucrose molecule to the C1 position of the fructose group of another sucrose molecule, resulting in the formation of 1-kestose (Fig. 5). Subsequently, 1-FFT extends the carbon chain based on sucrose, leading to the formation of inulin-type fructans with different degrees of polymerization in a reversible one-step reaction (*Márquez-López, Loyola-Vargas & Santiago-García, 2022*). The spatial and temporal distribution of fructans, similar to other polysaccharides, is determined by the balance between biosynthesis and degradation pathways. Fructan homeostasis, in turn, is affected by changes in plant growth stages and environmental conditions.

## Regulation mechanism of tuber plant polysaccharide enzyme

The biosynthesis of polysaccharides is regulated by a variety of enzymes encoded by structural genes, including SUS, SPS, INV, HXK, FRK, UGPase, and GTs (as shown in Table 2, Fig. 1). Sucrose decomposition is primarily controlled by SUS and INV. SUS facilitates the reversible cleavage of sucrose into fructose and UDP-glucose or adenosine diphosphate glucose, while SPS converts UDP-glucose into sucrose (*Boulanger et al., 2021*). In contrast, INV plays a role in breaking down sucrose into fructose and glucose in

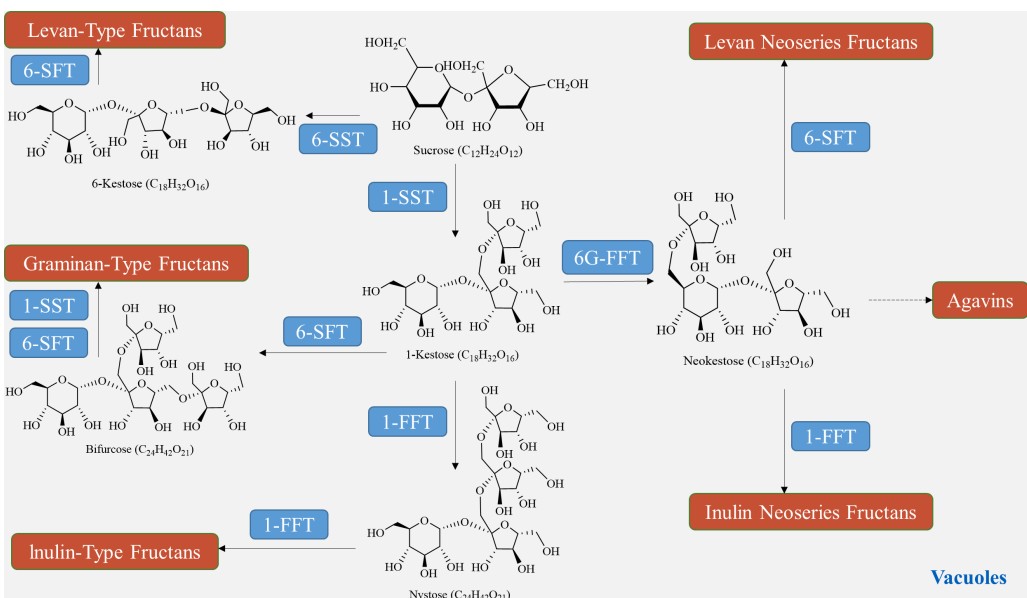

**Figure 5 Fructan biosynthesis pathways.**

plants. It is involved in cell osmotic regulation and the accumulation of sugar in storage organs, providing substrates for the subsequent synthesis of polysaccharides (*Chen et al., 2022*). HXK facilitates glucose phosphorylation in plant cells, while fructose undergoes phosphorylation by fructose kinase prior to metabolism. The second part involves the biosynthetic pathway from sucrose to GDP-mannose and GDP-fucose, as described by *Sosicka et al. (2022)*. Firstly, SUS catalyzes the conversion of sucrose to fructose, which is then further converted into fructose-6-phosphate with the help of HXK (*Lehretz et al., 2020*). Subsequently, fructose-6-phosphate is transformed into mannose-6-phosphate through the catalysis of mannose-6-phosphate isomerase (MPI). This is followed by the conversion of mannose-6-phosphate into mannose-1-phosphate by phosphomannose isomerase (PMM), ultimately leading to the formation of GDP-mannose with the action of GDP-mannose pyrophosphorylase (GMPP) (*Fang et al., 2022*). GDP-mannose can be utilized as a precursor for synthesizing various compounds such as UDP-rhamnose, UDP-fucose, UDP-galactose, UDP-xylose, and others (*Beerens, Gevaert & Desmet, 2021*). UDP-glucose can directly convert to UDP-galactose with the help of UDP-glucose-4-epimerase (UGE) (*Althammer et al., 2022*). UDP-glucose hydrogenase is also able to catalyze the conversion of UDP-glucose into UDP-glucuronic acid, which serves as a precursor for synthesizing hyaluronic acid on the plasma membrane, as well as UDP-xylose and proteoglycan synthesis in the Golgi apparatus (*Zimmer, Barycki & Simpson, 2021*). Importantly, the conversion process between UDP-glucuronic acid and UDP-galacturonic acid by UDP-glucuronic acid isomerase further increases the complexity of polysaccharide composition.

Peer**J**

**Table 2  Functions of key enzymes in polysaccharide synthesis.**

| Key enzyme | Role in carbohydrate synthesis | Other physiological functions | Influencing factors of activity | Reference |
|---|---|---|---|---|
| Sucrose synthase (SUS) | It catalyzes the decomposition and synthesis of sucrose, that is, sucrose is decomposed into UDP-glucose and fructose in the presence of UDP, and its reverse reaction | Sucrose is transported to multiple pathways to provide the precursor UDP-glucose for the biosynthesis of cell wall polymers and starch; It helps the growth, development and metabolism of sink organs, and help plants adapt to abiotic stress environments such as hypoxia and cold | Fructose and UDPG inhibited degradation activity, UDP inhibited synthesis activity, and glucose inhibited synthesis and degradation; inhibited by $Zn^{2+}$, $Hg^{2+}$, $Gu^{2+}$, $Fe^{2+}$, $Ni^{2+}$ and $Co^{2+}$ | *Sheng et al. (2023)* |
| Sucrose invertase (INV) | Irreversibly catalyze the hydrolysis of sucrose into glucose and fructose | It regulates the distribution and utilization of assimilates in the sink organ, for example, in the sink tissue, the cell wall sucrose invertase hydrolyzes sucrose into hexose, and the hexoglycoprotein transports hexose into the cells of the sink tissue, thereby reducing the sucrose concentration outside the sink tissue and driving the extracellular phloem of the sink organ to unload; delaying leaf senescence; early defense in disease resistance and plant-symbiont system | By transcription, translation, post-translational modification; induced by low temperature, stress and maturation | *Coculo & Lionetti (2022)* |
| Sucrose phosphateSynthase (SPS) | SPS catalyzes the conversion of UDPG and fructose-6-phosphate (F6P) to sucrose-6-phosphate (S6P) in plants. In the ripening stage of fruit, the expression and activity of SPS were up-regulated, which promoted the synthesis of sucrose and increased the sweetness of fruit | By regulating sucrose synthesis to cope with cold, drought, water shortage and other environmental stress pressure | Phosphate and sucrose-6-phosphate are SPS inhibitors. Fructose-6-phosphate and 1,5-anhydroglucitol-6-phosphate could activate SPS activity. Histidine participates in the catalytic reaction; under the regulation of post-translational modification, SPS is phosphorylated in the dark and its activity is inhibited. Under light, SPS is phosphorylated and its activity is restored, indicating that phosphorylation and dephosphorylation can regulate the activity of SPS in plants | *Liao et al. (2022)* |
| Hexokinase (HXK) | Phosphorylation of several hexoses, including d-glucose (Glc), d-fructose (Fru), d-mannose (Man) and d-galactose (Gal) | as a hexose sensor, it plays a variety of roles in regulating plant growth, sugar sensors, regulating sugar signal transduction, and cooperating with plant hormones | Glucose excess can cause HXK sugar sensing pathway, T6P pathway and rapamycin (TOR) kinase target pathway to play a role. | *Dou et al. (2022)* |
| Fructokinase (FRK) | Phosphorylation of free fructose with high substrate specificity and affinity | It is specifically expressed in the anther during the late stage of pollen development and pollen germination, and regulates the acquisition of carbohydrates required for cell wall synthesis during pollen development; plays a role in long-term developmental processes in vascular development. | Only regulated by its own substrate fructose | *Fan et al. (2022)* |
| UDP-glucose pyrophosphorylase (UGPase) | UDPG was synthesized by the reaction of glucose 1-phosphate with UTP | Involved in carbohydrate metabolism, cell wall biosynthesis and protein glycosylation metabolism; regulating plant cell apoptosis in chloroplasts | It is regulated by Suc (the major transport form of carbon in plants). It was strongly up-regulated by low temperature and down-regulated by drought and flood conditions. | *Xu et al. (2022)* |

Xu et al. (2024), *PeerJ*, DOI 10.7717/peerj.17052

**Table 2** (*continued*)

| Key enzyme | Role in carbohydrate synthesis | Other physiological functions | Influencing factors of activity | Reference |
|---|---|---|---|---|
| Glycosyl trans-ferases (GTs) | Catalytically activated glycosyl donors are transferred to specific receptor molecules to form glycosidic bonds | The glycosylation process can increase the polarity and water solubility of antibiotics, so that they can reach the ideal effective concentration inside or outside the cell, and the presence of glycosylation can also enhance the chemical stability of antibiotics. Glycosylation can specifically recognize biological targets and play a key role in bacteriostasis. | N-glycosylation strongly affects GTs activity and Golgi localization. | *Kurze et al. (2022)* |

## Hexokinase-regulated polysaccharide synthesis mechanism

The first glucose sensor discovered in plants, hexokinase (HXK), is now recognized as a dual-functional protein that performs important roles in plant carbohydrate metabolism. HXK activity is tightly regulated by various internal and external signals, including sugar availability, hormonal signals, and environmental stresses. In plant cells, hexokinase has been shown to directly regulate starch synthesis through the up-regulation of starch synthesis genes under sugar deprivation conditions (*Bouwman et al., 2020*). Additionally, it indirectly affects the expression of genes involved in cellulose biosynthesis (*Hoffmann et al., 2021*). The regulatory mechanisms of hexokinase in plant polysaccharide synthesis are complex, involving both direct and indirect mechanisms that govern the process. It is expected that new discoveries in this field will enhance our understanding of these mechanisms and offer valuable insights into innovative strategies for manipulating plant polysaccharide synthesis in biotechnological applications.

Hexokinase-mediated glucose and fructose phosphorylation initiates intracellular metabolism by stimulating glycolysis to produce secondary metabolites and energy. Before entering the cytoplasm, HXK, which is located on the outer membrane of the chloroplast, catalyzes the conversion of glucose to glucose-6-phosphate (G6P) (*Sonagra & Motiani, 2022*). G6P acts as the intersection point between glycolysis and the pentose phosphate pathway (Fig. 1), where G6P dehydrogenase further catalyzes its dehydrogenation to NADPH (*Laporte, González & Moenne, 2020*). HXK regulates and provides substrates for various pathways, including starch synthesis, fatty acid synthesis, nucleotide formation, and the oxidative pentose phosphate pathway (OPPP) (*Ren et al., 2022*). Moreover, glucose alone can activate root meristem through the TOR signaling pathway during the transition from heterotrophic to autotrophic metabolism in *Arabidopsis*. This activation is dependent on glucose phosphorylation by HXK, which supplies intermediate metabolites for cell wall synthesis and the energy needed for meristem proliferation (*Ye et al., 2022*). Additionally, hexose monophosphate within the cytosol serves as the substrate for nucleoside diphosphate sugar synthesis in cell wall biosynthesis. The inhibition of HXK by competitive inhibitors also leads to the inhibition of polysaccharide biosynthesis. Moreover, the glucose-hexose kinase system is involved in various forms of abiotic stress, such as salt tolerance, osmotic stress tolerance, and anthocyanin accumulation (*Aziz & Mohiuddin, 2023*).

HXK not only participates in glucose phosphorylation but also contributes to the regulation of some photosynthetic genes. As a sugar detecting protein, HXK is engaged in sugar signal transduction, which involves sensing stress, light, hormones, and nutrients, therefore influencing gene expression, as well as the growth and development of plants. HXK exists as a multigene family in plants, and there are 7 *MeHXK*s in cassava (*Lai et al., 2022*). At now, GenBank contains a compilation of HXK homologous genes from 28 different higher plants. The HXK gene family predominantly consists of 9 exons and codes for 492–522 amino acids (*Dou et al., 2022*). The subcellular localization analysis of HXK demonstrates that inside plant cells, the majority of HXK family members are mostly found in mitochondria, while only a small number are present in the cytoplasm, chloroplasts, and plastid matrix (*Chen, Tian & Guo, 2023*). The majority of the HXK gene family members exhibit expression in various organs or tissues, although *Arabidopsis thaliana AtHKL3* and

*Oryza sativa OsHXK10* are solely expressed in flowers (*Zheng et al., 2020*). The metabolic processes of sucrose and its subsequent breakdown into hexoses play a crucial role in controlling the storage and utilization of carbon, as well as other signaling pathways related to sugar. The intricate metabolic processes work together to uphold the balance of energy within cells and guarantee the organized progression of biochemical reactions in plants.

## Glycosyltransferase-catalyzed polysaccharide synthesis mechanism

Glycosyl transferases (GTs) are essential enzymes involved in glycosylation and are pivotal in the creation of glycosidic linkages. The control of polysaccharide production in plants is achieved by the regulation of GTs. This regulation encompasses various processes, including gene expression, protein modification, and enzyme activity regulation (*Tan et al., 2023*). The availability and activity of nucleotide sugar donors, which are generated in the cytosol and transferred to the Golgi apparatus, significantly affect GTs activity. These donors are used by GTs to create polysaccharides. Within the realm of plants, multiple transcription factors have been recognized as crucial controllers of GTs gene expression (*Lu et al., 2023*). These transcription factors selectively attach to distinct DNA sequences located in the promoter regions of GTs genes, and they have the ability to either stimulate or inhibit the expression of these genes. Furthermore, GTs activity can be influenced by post-translational changes such as phosphorylation, acetylation, and glycosylation, which can either enhance or limit the enzyme's activity. Environmental factors such as temperature, light, and nutrition availability also influence the control of GTs activity and the production of polysaccharides (*Tan et al., 2023*). Nucleotide sugar transporters facilitate the transportation of UDP monosaccharides and GDP monosaccharides from the cytoplasm of plants to the Golgi apparatus. Through the action of GTs, monosaccharide residues are moved from active nucleotide sugars to elongated polysaccharide chains (*Lu et al., 2023*). This process involves dehydration and condensation, resulting in the formation of polysaccharides. These polysaccharides are subsequently transported to various plant regions for storage using secretory vesicles (*Zhang et al., 2023*).

Glycosylation is vital for the production of secondary metabolites. GTs play a crucial role in transferring sugar groups from donor molecules to receptor molecules, leading to the creation of different glycoside chemicals (Fig. 1). There are two basic forms of GTs that play a role in polysaccharide biosynthesis: one has a single transmembrane domain, while the other has numerous transmembrane domains (*Zabotina, Zhang & Weerts, 2021*). Annotating GT family genes associated with plant polysaccharide synthesis has been performed in the CAZY database (http://www.cazy.org/) (*Tan et al., 2023*). Within this group, the GT1 family has the highest number of members and the most varied range of functions. The GT1 family, referred to as UDP-glycosyltransferases (UGT), predominantly facilitates the transfer of uridine diphosphate (UDP) sugars to particular receptors, including proteins, antibiotics, nucleic acids, plant hormones, and other substances (*Lu et al., 2023*).

The catalytic processes of glycosyltransferases can be categorized into two groups, retention and inversion, based on the stereochemical heterogeneity of glycosylation substrates and products (Fig. 6). The configurational inversion catalytic process entails

**Figure 6** **Method of glycosyltransferase catalyzing monomer transfer.**

the active hydrogen of the acceptor molecule being captured by the basic amino acid of the GTs. Subsequently, the oxygen anion attacks the acceptor molecule's carbon from the opposite side of the donor molecule, resulting in the formation of an oxocarbenium ion-like transition state (*Wang et al., 2019*). After the phosphate group is removed, the catalytic process of configurational inversion is finished, matching the mechanism used by glycoside hydrolases to break glycosidic bonds (*Sun et al., 2021*). The precise mechanism of the configuration-preserving type has not been completely comprehended. The initial proposal of the double-replacement process for the creation of glycosyl-enzyme covalent intermediates was substantiated by employing chemical interventions with sodium azide to rectify a mutated variant of α3-galactosyltransferase (α3GalT) (*Franceus & Desmet, 2020*). In the absence of definitive experimental evidence supporting alternative covalent intermediates, the SNi 'internal return' catalytic mechanism was also suggested. This mechanism suggests that the nucleophilic attack and departure of the leaving group take place on the same side of the sugar group, resulting in the formation of a brief oxocarbenium-like transition state. Subsequently, the acceptor C-O glycosidic link is formed and the C-O bond between the sugar group and the phosphate group is cleaved, so concluding the catalytic process of configuration retention (*Mahajan et al., 2021*). Ultimately, additional experimental evidence is required to determine whether the configuration-preserving GTs may be executed using various catalytic methods.

## PROSPECT

### Further exploration of polysaccharide synthesis mechanism in tuber plants

The progressive advancement of detection techniques and analytical equipment has facilitated thorough examination of the polysaccharide production pathway in several metabolic pathways of tuber plants. This study provides a comprehensive overview of the genetic mechanisms governing the production of starch, cellulose, pectin, and fructan in tuber plants. Additionally, it examines the catalytic mechanism of crucial enzymes involved in these processes. This overview provides a fundamental basis for future research on the control of plant polysaccharide synthesis, which may lead to the advancement of novel technologies and methodologies for the production and application of these vital chemicals.

The continuous development of detection techniques and analytical equipment has enabled comprehensive investigation of the polysaccharide synthesis pathway in several metabolic pathways of tuber plants. This study offers a thorough examination of the genetic pathways that regulate the synthesis of starch, cellulose, pectin, and fructan in tuberous plants. Furthermore, it investigates the catalytic mechanism of essential enzymes engaged in these activities. This overview serves as a foundational framework for future study on the regulation of plant polysaccharide synthesis, potentially resulting in the development of innovative technologies and approaches for the production and utilization of these essential compounds.

Studies in tuber plants of potatoes (*Odgerel & Banfalvi, 2021*), yams, and taro (*Rinaldo, 2020*), have demonstrated that heterologous expression, *in vitro* enzymatic catalysis, gene knockout, and RNA interference techniques can clarify the mechanism and catalytic activity of key enzymes in polysaccharide synthesis. These techniques provide a foundation for the directional synthesis of target polysaccharides in tuber plants (*Tang et al., 2022*).

Further research could focus on examining the impact of epigenetic pathways on the regulation of polysaccharide biosynthesis. Epigenetic alterations, such as DNA methylation and histone modification, have been demonstrated to have pivotal functions in governing gene expression. Hence, understanding the significance of epigenetic control in this mechanism could pave the way for the creation of innovative technologies to enhance the synthesis of these crucial molecules.

Subsequent investigations could prioritize the analysis of how epigenetic mechanisms affect the control of polysaccharide production. Epigenetic changes, such as DNA methylation and histone modification, play a crucial role in controlling gene expression. Therefore, comprehending the importance of epigenetic regulation in this process could lead to the development of cutting-edge technologies to improve the production of these vital compounds.

Environmental and developmental variables exert an influence on the manufacture of polysaccharides in tuber plants. Plant hormones, such as gibberellins (GA) and abscisic acid (ABA), are known to have important functions in controlling plant growth and development. Gaining insight into the indirect impact of these factors on polysaccharide production could pave the way for innovative strategies to optimize their production.
Polysaccharides are multifunctional substances that have a wide array of uses in the fields of food and beverage manufacturing, pharmaceutical research, and material engineering. Enhanced comprehension of the genetics underlying polysaccharide biosynthesis in tuber plants may facilitate the formulation of novel approaches to enhance their production for targeted purposes. Increasing the production of starch or cellulose can result in the creation of innovative food products. Improving the synthesis of polysaccharides with targeted characteristics can be beneficial for pharmaceutical research and development.

In summary, identifying the areas for future research will assist in filling the knowledge gaps about the genetic pathways involved in polysaccharide production in tuber plants. These study fields have the potential to reveal new possibilities for the manufacture and exploitation of these significant chemicals..

## Revelatory effects from the synthesis of other glycoside metabolites

Glycosides are the main form in which plant flavonoid metabolites are often present. Polysaccharides, being glycosides, are capable of undergoing hydrolysis. UGT enzymes can alter the glycosylation process of flavonoids, resulting in the formation of diverse flavonoid glycosides. An example of this is the upregulation of the coding gene *Gm UGT88A13* in soybean, which led to a large rise in the levels of isoflavones and flavonol glycosides in soybean hairy roots (*Johny et al., 2020*). Another instance is the soybean *Gm UGT79A6* gene, which produces an enzyme called flavonol 3-O-glucoside (1–6) rhamnosyl transferase. This enzyme enhances the amount of kaempferol 3-O-rutinoside in immature soybean leaves by attaching sugar molecules to flavonols. UGTs have the ability to utilize UDP-rhamnose as a source of sugar to produce flavonoid glycosides within living organisms (*Odgerel & Banfalvi, 2021*). The Rice *GSA1* gene has increased levels of glycosyltransferase activity specifically for flavonoids and lignans. The overexpression of *GSA1* leads to a modification in the flavonoid content, which subsequently influences the auxin level and the expression of genes connected to it in rice (*Dong et al., 2020*). The UDP-glycosyltransferase *CsUGT85A53* from *Camellia sinensis* has the ability to add a glucose molecule to ABA, resulting in the formation of an inactive ABA-glycoside. This process occurs both in laboratory conditions (in vitro) and in living plants (in planta) (*Jing et al., 2020*).

Glycosylation is the final stage in the production of flavonoid glycosides, altering the polarity of flavonoid molecules. Furthermore, it impacts the pharmacodynamic activity and pharmacokinetics of flavonoids. GTs offer notable benefits, including precise control over both the region and stereochemistry, as well as the ability to produce glycosidic linkages in high yields. UGTs have the ability to facilitate the addition of sugar molecules to terpenoids through a process called glycosylation. Presently, there is a significant focus on the examination of terpenoids, including nerol, linalool, steviol, and saponins. An example of this is that the transcription level of *UGT85A84* in *Osmanthus fragrans* is directly related to the accumulation of glycosides. The UGT85A84 protein variant has the ability to facilitate the glycosylation process of linalool and linalool oxide, resulting in the formation of a glycosylation conjugate of the aromatic molecule (*Fu et al., 2022*). Glycosylation of these molecules facilitates the investigation of information inside plant

polysaccharide production pathways and establishes a basis for comprehending the intricate genetic systems of plants. Furthermore, it offers assistance in investigating the principles governing plant life phenomena and examining the fundamental factors contributing to the occurrence and progression of life.

## CONCLUSIONS

Polysaccharides play a crucial role in various functions within tuber plants, including energy storage, structural support, and protection against pathogens and environmental stresses. The synthesis of these compounds begins with the conversion of photosynthetic sugar metabolites. As plants undergo photosynthesis, they produce sugars from carbon dioxide and water, which are then present in the form of starch, cellulose, and pectin in plant tubers. These sugars are subsequently broken down to create precursor compounds that selectively enter the polysaccharide synthesis pathway. Key enzymes, such as SUS, INV, UGPase, FRK, and GTs, are essential in the process of polysaccharide polymerization. The genetic mechanisms involved in tuber plant polysaccharide biosynthesis includes the activity of enzymes responsible for polysaccharide synthesis, such as cellulose synthase, starch synthase, and glycosyltransferases, which transfer sugars between donor and acceptor molecules. This article provides a summary of the biosynthesis pathway of polysaccharides and the role of key enzymes in tuber plants.

Comprehending the molecular process of polysaccharide synthesis in tuber plants not only aids in understanding how important enzymes in polysaccharide synthesis are regulated, but also serves as a theoretical basis for regulating metabolite synthesis in plant cells. By integrating the biosynthetic traits, spatial distribution, and sensing capabilities of tuber plant polysaccharides with other metabolic pathways, it is feasible to enhance plant productivity and manufacture certain polysaccharides in a controlled manner. The exploration of novel techniques for synthesizing and harnessing polysaccharides in tuber plants holds great promise for various industries, including food and beverage manufacturing, pharmaceutical research, and the creation of innovative materials.

### Funding

This research was supported financially by the National Natural Science Foundation of China (32260089), the Science and Technology Department Foundation of Guizhou Province of China (QKPTRC[2019]-027), the Special Joint Bidding Project of Zunyi Sci & Tech Bureau and Zunyi Medical University (ZSKHHZ-2020-91), the Graduate Education and Teaching Perform Project of Zunyi Medical University (ZYK105), the Research on Innovation and Entrepreneurship Education of Guizhou Ordinary Undergraduate Colleges (2022SCJZW10) and the Future Outstanding Teachers Training Program of Zunyi Medical University (XJ2023-JX-01-06). The funders had no role in study design, data collection and analysis, decision to publish, or preparation of the manuscript.

## Grant Disclosures

The following grant information was disclosed by the authors:

The National Natural Science Foundation of China: 32260089.

the Science and Technology Department Foundation of Guizhou Province of China: QKPTRC[2019]-027.

Special Joint Bidding Project of Zunyi Sci & Tech Bureau and Zunyi Medical University: ZSKHHZ-2020-91.

Graduate Education and Teaching Perform Project of Zunyi Medical University: ZYK105.

Research on Innovation and Entrepreneurship Education of Guizhou Ordinary Undergraduate Colleges: 2022SCJZW10.

Future Outstanding Teachers Training Program of Zunyi Medical University: XJ2023-JX-01-06.

## Competing Interests

The authors declare there are no competing interests.

## Author Contributions

- Mengwei Xu conceived and designed the experiments, performed the experiments, analyzed the data, prepared figures and/or tables, authored or reviewed drafts of the article, and approved the final draft.
- Jiao Hu performed the experiments, prepared figures and/or tables, and approved the final draft.
- Hongwei Li performed the experiments, authored or reviewed drafts of the article, and approved the final draft.
- Kunqian Li performed the experiments, authored or reviewed drafts of the article, and approved the final draft.
- Delin Xu conceived and designed the experiments, performed the experiments, prepared figures and/or tables, authored or reviewed drafts of the article, and approved the final draft.

## Data Availability

This is a literature review.

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
