# Peer review of "Research overview on the genetic mechanism underlying the biosynthesis of polysaccharide in tuber plants"

_PeerJ, doi:10.7717/peerj.17052_

## Round 0.1 · original submission · Major Revisions

Please revise according to the reviewer's comments and pay attention to the differences between this paper and the existing reviews.

Reviewer 1 ·

Basic reporting

The content of this review is in line with the scope of the journal, the topic of the review is innovative to a certain extent, and the introduction summarizes the research background and significance of this paper. However, the language of the paper needs to be modified, especially the professional vocabulary about carbohydrates.

Experimental design

Literature research is sufficient, the paper has a certain logic.

Validity of the findings

The conclusion is basically reasonable.

Additional comments

1.Line 12-13, This statement is not appropriate. Tuber plants and rice and wheat are not parallel words. Besides, corn is the number one food crop in the world.
2.Line 36, “glycogen” is a polysaccharide that comes from animals and is not appropriate in this article.
3.Line 43-46, The logical relationship here is weak. The authors need to explain why understanding the synthesis mechanism of polysaccharides is helpful for the application of polysaccharides.
4.Line 58-60, The main differences between the existing review and the published review should be explained in detail to illustrate the innovation of this paper.
5.Line 77-86, “The applications of polysaccharides can be divided into two categories.”The application division of the two aspects is not comprehensive, in addition to the polysaccharide as a medical material and functional factors, in fact, the content of the vast majority of the polysaccharide is starch, starch has no special health care effect, only as an energy substance.
6.As for the synthesis mechanism of various polysaccharides, this maunscript mainly focuses on the role of enzymes. However, the environment and plant nutrition conditions also affect the rationality of polysaccharides, and the author should add this discussion.
7.Line 170-286, “Pectin biosynthesis pathway”, Methyl ester group and acetyl group are important components in the molecular structure of pectin. The synthesis of these groups should be explained in this manuscript.

Reviewer 2 ·

Basic reporting

In this manuscript, the authors have comprehensively summarized the biosynthesis pathway of tuber plant polysaccharides. The data were rich and the figures were well-prepared. From my point of view, there were several comments for further improving the manuscript.

Experimental design

1. Since starch, cellulose, and pectin are regular and simple polysaccharides, the biosynthesis of these polysaccharides may lack novelty as a review paper. The introduction of complex polysaccharides may be more attractive, for instance, Bletilla striata polysaccharides.
2. More frontier questions for polysaccharide biosynthesis could be addressed. The presentative contribution of scholars should better be pointed out.
3. A comparison of simple polysaccharides and complex polysaccharides could be more interesting. The challenges for exploring polysaccharide biosynthesis should better be highlighted.

Validity of the findings

4. It would be more meaningful to point out the significance of polysaccharide biosynthesis in plants for biomedical industry or drug development.

Additional comments

NO

---

## Round 0.2 · Minor Revisions

Please revise as the reviewer‘s comment. In addition, I suggest the authors optimize the language of the whole paper, and the background color of the Figs ican be removed to see if it suitable for reading.

**Language Note:** The Academic Editor has identified that the English language must be improved. PeerJ can provide language editing services - please contact us at copyediting@peerj.com for pricing (be sure to provide your manuscript number and title). Alternatively, you should make your own arrangements to improve the language quality and provide details in your response letter. – PeerJ Staff

Reviewer 1 ·

Basic reporting

no comment

Experimental design

no comment

Validity of the findings

no comment

Additional comments

Thank you for your revision of the manuscript.

Reviewer 2 ·

Basic reporting

The author has well explained the three comments mentioned before. However, the last comment was not replied to. Please check and supplement.

Experimental design

No comment.

Validity of the findings

4. It would be more meaningful to point out the significance of polysaccharide biosynthesis in plant for biomedical industry or drug development.

Additional comments

No comment

---

## Round 0.3 · accepted · Accept

The reviewers and my questions were properly answered and this article can be accepted, congratulations!